# PEG-BHD1028 Peptide Regulates Insulin Resistance and Fatty Acid β-Oxidation, and Mitochondrial Biogenesis by Binding to Two Heterogeneous Binding Sites of Adiponectin Receptors, AdipoR1 and AdipoR2

**DOI:** 10.3390/ijms22020884

**Published:** 2021-01-17

**Authors:** In Kyung Lee, Gyuyoup Kim, Do-Hwi Kim, Brian B. Kim

**Affiliations:** EncuraGen, Inc., Anyang, Gyeonggi-do 14057, Korea; pobh33@encuragen.com (I.K.L.); alex@encuragen.com (G.K.); dhkim@encuragen.com (D.-H.K.)

**Keywords:** PEG-BHD1028, adiponectin, peptide drug, AdipoR1, AdipoR2, insulin resistance, glucose uptake, mitochondrial biogenesis, fatty acid β-oxidation, glucose tolerance

## Abstract

Adiponectin plays multiple critical roles in modulating various physiological processes by binding to its receptors. The functions of PEG-BHD1028, a potent novel peptide agonist to AdipoRs, was evaluated using in vitro and in vivo models based on the reported action spectrum of adiponectin. To confirm the design concept of PEG-BHD1028, the binding sites and their affinities were analyzed using the SPR (Surface Plasmon Resonance) assay. The results revealed that PEG-BHD1028 was bound to two heterogeneous binding sites of AdipoR1 and AdipoR2 with a relatively high affinity. In C2C12 cells, PEG-BHD1028 significantly activated AMPK and subsequent pathways and enhanced fatty acid β-oxidation and mitochondrial biogenesis. Furthermore, it also facilitated glucose uptake by lowering insulin resistance in insulin-resistant C2C12 cells. PEG-BHD1028 significantly reduced the fasting plasma glucose level in *db/db* mice following a single s.c. injection of 50, 100, and 200 μg/Kg and glucose tolerance at a dose of 50 μg/Kg with significantly decreased insulin production. The animals received 5, 25, and 50 μg/Kg of PEG-BHD1028 for 21 days significantly lost their weight after 18 days in a range of 5–7%. These results imply the development of PEG-BHD1028 as a potential adiponectin replacement therapeutic agent.

## 1. Introduction

Drugs exert their functions by binding to specific receptors and subsequently modulating disease-associated protein interactions. Among pharmaceutical compounds, peptides exhibit unique therapeutical and biochemical characteristics positioning between small molecules and proteins [1,2]. Although both peptides and proteins are superior in binding large, flat, and hydrophobic binding interfaces, the most common biochemical nature of the receptor proteins, relatively larger and more complex forms make proteins more challenging to penetrate the cell membrane than peptides [3,4]. Advanced modern technologies also allowed the design of a peptide for challenging binding interfaces with a high affinity and specificity [1,5]. In addition, the physicochemical properties of peptides are often chemically modified with various methodologies such as conjugation with large polymer, fusion to long-lived proteins, and amino acid substitutions to overcome inherited challenges associated with peptide drugs like instabilities and a short half-life (e.g., 2–3 min) in the systemic blood circulation [6]. More than 60 peptide drugs are currently approved in the United States and other regions, and approximately 20 peptide drugs annually enter into clinical development programs [7]. Several companies are developing GLP-1 based multifunctional peptides such as GLP-1/GIP dual peptide and GLP-1/GLG dual peptide [8]. Several adiponectin-receptor agonist peptides like ADP355, Pep 72, and Pep77 have been recently introduced as potential adiponectin replacement therapeutic agents [9,10]. Among these molecules, ADP355 is currently at the human clinical trial stage for the dry eye syndrome indication [11].

The author and his colleagues have also recently introduced BHD1028 (1928 Dalton of molecular weight), a novel peptide agonist to AdipoR1. The design of the peptide reflected novel findings, including the presence of two heterogeneous binding sites on the receptor, and the peptide bound to AdipoR1 with a higher affinity than the peptide binding to one binding site on the receptor [12]. Since adiponectin exerts its physiological regulatory functions by binding to its receptors as a multimeric form and activating subsequent signal pathways, the binding affinity to the receptor(s) with the right configuration could be of the critical parameters in developing an appropriate agonist to the receptor(s). Despite the high binding affinity to AdipoR1, BHD1028 exhibited poor solubility in an aqueous solvent because of its hydrophobic nature. The insolubility was significantly improved by conjugating 5000 Dalton polyethylene glycol (PEG) to the N-terminal of the molecule, designated as PEG-BHD1028 (6928 Dalton). In this study, based on the information of the potential presence of heterogeneous dual binding sites [12] and approximately 80% of structural homology between AdipoR1 and AdipoR2 without N-terminal specific amino acid sequences [13], the binding affinity of BHD1028 to both receptors was examined using the SPR (surface plasmon resonance) technology. Additionally, the critical biochemical and biological mode of actions of PEG-BHD1028 was evaluated in comparison with the reported activities of adiponectin using in vitro and in vivo testing models.

## 2. Results

### 2.1. Determination of Binding Characteristics between BHD1028 and AdipoRs

The efficacy of a drug targeting a receptor is dependent on its binding affinity and effective activation of desired biological processes inside cells. Although BHD1028 was initially designed as an AdipoR1 agonist [12], it was expected to bind to both AdipoR1 and AdipoR2 based on an approximately 80% structural homology between two receptors except for the N-terminal specific sequence located in the cytoplasmic region [13]. The design also reflected the potential presence of heterogeneous dual binding sites on each receptor [12]. In this study, the intermolecular binding affinity between BHD1028 and AdipoRs and the affinity to each of the two binding sites within a receptor were evaluated using a surface plasmon resonance (SPR) assay. The results indicated that BHD1028 bound to two different binding sites on each receptor with different binding affinities. It bound to AdipoR1 with a higher affinity than AdipoR2 at KD (equilibrium dissociation rate constant) of 7.38 and 11.26 μM, respectively, at the binding site 1 and 36.55 and 25.06 μM, respectively, at the binding site 2 (Table 1). The binding affinity to binding site 2 was relatively lower than binding site 1. These results represent that BHD1028 has a high binding affinity to both AdipoR1 and AdipoR2 receptors, but the affinity to binding site 1 was approximately 2 to 5 folds greater than binding site 2 on each receptor. The affinity of BHD1028 was tested at more than 5 concentrations, and chi-squares were less than 20, confirming that the tests were conducted under appropriate test conditions (Figure 1).

### 2.2. The Effect of PEG-BHD1028 on Cell Viability in C2C12 Myotubes

A cell viability test was performed using C2C12 cells to assure that the concentrations applied to this study are adequate for functional assessment of PEG-BHD1028. The results showed no cytotoxic effect up to 1 mM (Figure 2).

### 2.3. The Effect of PEG-BHD1028 on the Phosphorylation of AMPK and ACC in C2C12 Myotubes

AMPK (AMP-activated protein kinase) has been known for its effects on regulating various metabolic pathways at the cellular level [14]. It is also known that adiponectin mainly exerts its function through the activation of the AMPK pathway, including the downstream ACC (Acetyl-CoA Carboxylase) signaling pathway, a crucial process for fatty acid biosynthesis and oxidation [14,15]. To evaluate the effect of PEG-BHD1028 on activating AMPK and ACC, C2C12 cells were treated with 10, 100, and 500 nM of PEG-BHD1028 and globular adiponectin (gAD) as a positive control, and the expressed proteins were analyzed using the western blot test (Figure 3A). As shown in Figure 3B,C, the phosphorylated AMPK at Thr172 and phosphorylated ACC following the treatment with PEG-BHD1028 significantly increased in a dose-dependent manner (*p* ≤ 0.05 and *p* ≤ 0.01). These results suggest that PEG-BHD1028 can significantly activate AMPK and ACC in C2C12 skeletal muscle cells.

### 2.4. Stimulation PPARα Transcription Activity by PEG-BHD1028

PPARα (Peroxisome proliferator-activated receptor alpha), also known as nuclear receptor subfamily 1, group C, member 1, is one of the key regulators in fatty acid metabolism. AdipoR2 primarily mediates its activation, although it is also expressed through the AMPK pathway mediated by AdipoR1 [16]. Activation of PPARα promotes uptake and catabolism of fatty acids by upregulating genes involved in fatty acid transport and fatty acid β-oxidation in peroxisome and mitochondria [17]. The expression of PPARα was examined in C2C12 myotubes following the treatment of 20, 100, and 500 nM of PEG-BHD1028. As shown in Figure 4, PEG-BHD1028 significantly increased the expression level of PPARα in a dose-dependent manner (*p* ≤ 0.05 at 500 nM and *p* ≤ 0.01 at 100 nM). Therefore, the results indicate PEG-BHD1028 effectively activates PPARα through binding to AdipoR1 and AdipoR2 implying that PEG-BHD1028 facilitates multiple, complex signaling pathways involved in energy metabolism, including AMPK, p38 MAPK, and PPARα similarly to the action mechanism of adiponectin.

### 2.5. Enhanced Mitochondrial Biogenesis Activities in C2C12 Skeletal Muscle Cells

Adiponectin has been reported to facilitate glucose utilization through the increased fatty acid β-oxidation resulted from enhanced mitochondrial biogenesis in peripheral tissues [18,19]. The process ultimately increases energy expenditure through elevated oxygen consumption and thermogenesis, resulting in weight loss [20]. Mitochondrial biogenesis is a critical cellular process in maintaining energy metabolism and constant energy supply by increasing mitochondrial mass, and a series of signal transductions are involved in the process. To assess the effect of PEG-BHD1028 on mitochondrial biogenesis, the expression levels of critical regulatory genes, such as p38 mitogen-activated protein kinases (p38 MAPK), MAP kinase phosphatase-1 (MKP-1), and peroxisome proliferator-activated receptor-gamma coactivator 1α (PGC-1α), were examined in C2C12 myotubes following the treatment of PEG-BHD1028. Figure 5A,B illustrate that PEG-BHD1028 significantly elevated the phosphorylation level of p38 MAPK, known to be a positive regulator of PGC-1α expression, in C2C12 cells at the concentration of 100 and 500 nM at *p* ≤ 0.01 and *p* ≤ 0.05, respectively. Additionally, it was also observed that PEG-BHD1028 significantly enhanced the expression of PGC-1α, an essential regulator for mitochondrial biogenesis (Figure 5C and Appendix A). By understanding the elevated expression of p38 MAPK and PGC-1α, we further assessed whether PEG-BHD1028 is involved in the regulatory process in metabolic homeostasis by examining the expression level of MAPK phosphatase-1 (MKP-1), a negative downstream regulator of p38 MAPK and PGC-1α in differentiated C2C12 cells [21]. The result showed that the expression level of MKP-1 was significantly decreased in a dose-dependent manner (Figure 5D). The elevated phosphorylation of p38 MAPK and PGC-1α and the decreased expression of MKP-1 after the treatment with PEG-BHD1028 clearly demonstrated that PEG-BHD1028 functioned in the resemblance of adiponectin in the regulation of metabolic homeostasis by inducing mitochondrial biogenesis.

Besides the verification of activated the signal cascade responsible for mitochondrial biogenesis, relative mitochondrial DNA contents were measured against nucleus DNA contents using a qPCR method. For this test, a ratio of the total DNA content between NADH dehydrogenase subunit 1 (ND1) in mitochondria and hexokinase in the nucleus was measured following the treatment of PEG-BHD1028. The number of the ND1 gene copies significantly increased relative to the total DNA of hexokinase following the PEG-BHD1028 and globular adiponectin treatment (Figure 6).

The effect of PEG-BHD1028 on the signal cascade of mitochondrial biogenesis led us to further examine its impact on the expression of mitochondrial genes. The expression of two mitochondrial genes, cytochrome c oxidase subunit 2 (COX2) and NADH dehydrogenase subunit 5 (ND5), was evaluated using a real-time PCR method. C2C12 cells were treated with 20, 100, and 500 nM of PEG-BHD1028. Significant elevation of COX2 and ND5 mRNA was observed at all concentrations, indicating an effective role of PEG-BHD1028 in enhancing mitochondrial gene expression as an adiponectin analog (Figure 7).

In addition to the effective facilitation of mitochondrial biogenesis and mitochondrial gene expression by PEG-BHD1028, activation of holistic mitochondrial activities was further verified using a fluorescence microscope. Mitochondrial fragmentation and intensity were measured using MitoTracker^TM^ following 24 h treatment of 20, 100, and 500 nM of PEG-BHD1028 (Figure 8A). Data showed significantly increased mitochondrial fragmentation and mitochondrial intensity compared to the negative control, indicating the enhancement of mitochondrial biogenesis (Figure 8B). These results suggest that PEG-BHD1028, like adiponectin, increases mitochondria activity and mitochondrial biogenesis.

### 2.6. PEG-BHD1028 Induces 2-Deoxy-D-glucose (2DG) Uptake in C2C12 Myotubes

Skeletal muscle is the primary site of glucose metabolism, accounting for approximately 35% of the entire body’s glucose uptake in the presence of insulin [22]. However, insulin-resistant tissue exhibits impaired glucose metabolism regardless of insulin. The effect of PEG-BHD1028 on glucose uptake in insulin-resistant C2C12 skeletal muscle cells was evaluated. To induce insulin resistance, differentiated C2C12 myotubes were treated with 0.6 mM palmitate and incubated for 16 h and then treated with 100 nM insulin and PEG-BHD1028 at concentrations of 100 and 500 nM for 30 min. In the non-resistant state, the treatment of 100 nM insulin alone significantly increased glucose uptake (Figure 9A), while glucose uptake was not observed in the insulin-resistant cells in the presence of insulin. The data showed that PEG-BHD1028 significantly increased glucose uptake in the insulin-resistant C2C12 myotubes in a dose-dependent manner (*p* ≤ 0.01) relative to untreated or insulin-only treated insulin-resistant cells (Figure 9B), implying that PEG-BHD1028 effectively ameliorates insulin resistance.

### 2.7. Glucose Lowering Effect and Profile of PEG-BHD1028

The pharmacodynamic profile of PEG-BHD1028 was evaluated in *db/db* mice following a single-dose s.c. injection of vehicle (control group) or 50, 100, and 200 μg/Kg of PEG-BHD1028 (experimental groups) under fasting conditions. Blood samples were collected hourly for the first 8 h and every 2 h for the following 4 h from the tail vein. The glucose levels of the experimental groups were significantly lower than the control group until 10 h either at *p* ≤ 0.05 or *p* ≤ 0.01, and remained relatively constant after 5 h (Figure 10). The magnitude of the glucose-lowering effect of 200 μg/Kg was greater than 50 or 100 μg/Kg for the first 2 to 3 h, but there was no noticeable difference after 3 h. The peak glucose-lowering effect of PEG-BHD1028 was observed between 4 and 6 h following the administration. These results indicate that PEG-BHD1028 enables to effectively decrease or regulate the blood glucose level without causing hypoglycemia.

### 2.8. Glucose Lowering Effect through the Amelioration of Insulin Resistance

An OGTT (Oral Glucose Tolerance Test) was performed to understand the mechanism of action of PEG-BHD1028 in the glucose-lowering effect using *db/db* mice, a type 2 diabetes animal model. Two groups (*n* = 5/group) of 6-h fasted animals subcutaneously received 1 and 50 μg/Kg of PEG-BHD1028 240 min before oral glucose loading at a dose of 1 g/Kg body weight. Blood samples were collected from the tail vein and tested for glucose and insulin concentrations at −240 min, 0 min, 15 min, and then every 30 min until 120 min. As shown in Figure 11A,B, the glucose level of the 50 μg/Kg received group was significantly lower than the control group (*p* ≤ 0.05) at 60, 90, and 120 min, and glucose AUC_1–120min_ of 50 μg/Kg group was significantly lower than that of the control group. However, the glucose-lowering effect of 1 μg/Kg was not significant. The insulin concentration of 50 μg/Kg dose group was also significantly lower than control animals at 60, 90, and 120 min (*p* ≤ 0.05 at 60 and 90 min; *p* ≤ 0.01 at 120 min) (Figure 11C). The insulin level of 50 μg/Kg group continuously decreased after the administration of the testing article, except for a brief increment in response to the glucose loading, until 120 min, while the insulin concentration of the control group remained relatively constant other than the elevation responding to the glucose-loading between 30 and 90 min. Although the insulin concentration of 1 μg/Kg group decreased relative to the time point 0, the difference was not significant. The insulin AUC_1–120min_ of 50 μg/Kg group was significantly lower than the control group (*p* ≤ 0.05) (Figure 11D). These results suggest that the significant glucose-lowering effect of PEG-BHD1028 is primarily through the amelioration of insulin resistance, not insulin secretion.

### 2.9. Weight Loss Effect of PEG-BHD1028 without Affecting Appetite

The effect of PEG-BHD1028 on weight loss and food intake was evaluated in *db/db* mice following the daily s.c. administration of 1, 5, 25, 50, and 200 μg/Kg of the testing article (experimental groups) and a vehicle for the control group for 3 weeks (*n* = 5/group). Regular rodent chow and drinking water were provided daily without diet restriction throughout the study. As shown in Figure 12A, there was no significant difference in food intake between groups during the study period. This finding represents that PEG-BHD1028 does not affect appetite (Appendix A) or other intrinsic factors that potentially influence appetite.

Interestingly, most of the animals in both groups gained or maintained weight for the initial 7 to 9 days after the administration of PEG-BHD1028, but the weight-gaining trend of the animals in the experimental group shifted to a negative-trend between days 9 and 12 (Figure 12B). One animal in the 5 μg/Kg group and two animals in the 200 μg/Kg group gained weight until day 15, and the weight of those animals remained relatively constant until the end of the study. Unlike the animals in 25 and 50 μg/Kg groups whose body weights turned to the negative trend from day 9, 3 animals in 200 μg/Kg group gained weight until day 12, and the gaining-trend turned to the negative from day 15 (Appendix A). The average body weight of the animals in 25 and 50 μg/Kg groups was significantly lower than the control animals from day 15 (*p* ≤ 0.05), and the loss-trend became more evident after day 15. On day 21, the animals in those groups lost approximately 4% and 6%, respectively, relative to day 0, and the difference was significant (*p* ≤ 0.01) compared to the control animals (Figure 12B). The average weight loss of the 5 μg/Kg group appeared to be greater than that of the other groups, but there was no significant difference from the control animals until day 19. This result may be due to a relatively wide weight variation within the group or small sample size. The average weight loss of 5 μg/Kg groups on day 21 was about 7% (*p* ≤ 0.05) relative to day 0.

## 3. Discussion

Numerous studies have been conducted to understand adiponectin and its receptors in various aspects, including detailed structural chemistry, physiological functions of in vitro and in vivo systems, and clinical significance since its first discovery in 1995. Although there are still ongoing endeavors to further understand the roles and mechanisms of action, a significant amount of scientific and clinical information has already revealed that adiponectin plays multiple beneficial roles in regulating physiological processes such as the amelioration of insulin resistance, enhancement of fatty acid β-oxidation, protection from cell apoptosis, and anti-inflammatory activity. Because of these extensive beneficial roles, adiponectin has drawn significant attention to developing it into a therapeutic agent for more effective treatment of the diseases associated with the hormone deficiency such as type 2 diabetes mellitus, obesity, and non-alcoholic steatohepatitis.

Several peptides such as ADP355, Pep72, and Pep77 have recently been introduced as an agonist to AdipoRs based on the various advanced technologies that allow to optimize the functionality of therapeutic peptides. PEG-BHD1028, a novel pegylated-peptide agonist to AdipoR1 and AdipoR2, was first introduced by Kim and his colleagues a couple of years ago [12]. It was designed as a mimetic of adiponectin reflecting the sequence of the active site of the hormone and two heterogeneous binding sites on each receptor. Ultimately, this molecule is expected to mimic the action mode of adiponectin.

In this study, PEG-BHD1028 was evaluated for its binding kinetics for AdipoR1 and AdipoR2 based on the design concept of the molecule and its ability to regulate biological processes using cell and animal models. In the surface plasmon resonance (SPR) assay utilizing human AdipoRs immobilized on the CM5 chip, BHD1028 exhibited tight binding to the binding domain of the receptors with a single or low two-digit micromolar level of KD (equilibrium dissociation rate constant). It is also expected that BHD1028 has a relatively higher affinity based on its dual binding configurations on each receptor, which may, in turn, lead to increased biological activities and, subsequently, clinical efficacy. Otvos also reported that the affinity of ADP399, two ADP355 conjugated-peptide, was approximately 20-folds higher relative to ADP355 [11]. These pieces of information may be crucial in the development of an effective agonist to AdipoRs. However, more extensive studies should be conducted to understand further how one or two heterogeneous binding configurations are related to its biological functions.

To delineate the functionality of PEG-BHD1028 as an adiponectin mimetic, we investigated the mechanism of action of PEG-BHD1028 on lipid metabolism and glucose homeostasis in cellular and animal models. It has been well established that activation of AdipoR1 activates the AMP-activated protein kinase (AMPK), the main entry of action on lipid metabolism. The phosphorylated AMPK further activates its subsequent downstream pathways such as the acetyl-CoA carboxylase (ACC) signal transduction, thereby inhibiting the formation of malonyl-CoA. The inhibition of malonyl-CoA ultimately promotes fatty acid β-oxidation. Recent studies also reported that adiponectin increases the quantity and function of mitochondria, a determinant of energy metabolism, through the activation of AMPK [23]. Activated AMPK and increased PGC-1α enhance muscle glucose metabolism by facilitating mitochondrial biogenesis [24]. In experiments using C2C12 cells, we have verified that PEG-BHD1028 activates the signal pathways responsible for fatty acid β-oxidation and mitochondrial biogenesis. Along with the elevated phosphorylated ACC, PEG-BHD1028 increased levels of the activated p38 mitogen-activated protein kinase (p38-MAPK), peroxisome proliferator-activated receptor alpha (PPARα), and PPARγ Coactivator-1α (PGC-1α), a multifunctional regulatory factor (Figure 13), indicating the significant effect of PEG-BHD1028 on lipid metabolism. These results are in alignment with the functions of adiponectin and represent opportunities for the intervention of various metabolic diseases.

It has also been well accepted that adiponectin alleviates insulin resistance, and adiponectin deficiency leads to abnormal glucose metabolism [25,26]. Thus, insulin resistance has been recognized as the most fundamental root cause of type 2 diabetes mellitus, and the concentration of circulating adiponectin is negatively correlated with insulin resistance. Hu et al. demonstrated that adiponectin enhanced glucose metabolism by improving insulin sensitivity on diabetic swine [27]. Recent studies also have shown that p38 MAPK acts as an essential mediator in regulating adiponectin-induced glucose uptake in response to insulin [28,29]. In this study, we observed that PEG-BHD1028 significantly increased 2-deoxyglucose uptake in insulin-resistance induced C2C12 skeletal muscle cells (Figure 9). Along with the OGTT result, this finding suggests that PEG-BHD1028 facilitates glucose uptake by reducing insulin resistance in the presence of insulin through the activation of glucose transporter (GLUT) following the activation of the AMPK/p38 MAPK signaling pathway. These features imply that PEG-BHD1028 could be a potential therapeutic agent for the treatment of type 2 diabetes mellitus.

X. D. Stepensky and his colleagues reported that the peak efficacy of metformin, a widely used insulin-sensitizing agent, was observed 4 to 6 h after the intraduodenal administration of 450 mg/Kg of metformin [30]. Despite the difference in pharmacodynamic profile due to the difference in pharmacokinetics and the proposed mechanism of action between two substances, the peak glucose-lowering effect of PEG-BHD1028 was also observed between 4 and 6 h after administration. Once the glucose level reached the lowest level after the administration of PEG-BHD1028, it remained relatively constant for 12 h without any sign of hypoglycemia. These pharmacodynamic profiles appeared to be different from insulin, which exerts its peak effectiveness in 2 to 3 h after treatment [31]. This difference may represent the difference in mechanism of action in glucose metabolism. The ability of PEG-BHD1028 to control glucose metabolism via the amelioration of insulin resistance was further demonstrated through the oral glucose tolerance test (OGTT). The glucose level significantly decreased in animals received 50 μg/Kg of PEG-BHD1028 relative to the control animals (*p* ≤ 0.05), and the level of insulin was significantly lower in the same animals than the control animals (*p* ≤ 0.01 and *p* ≤ 0.05). Along with the in vitro glucose uptake study results using insulin-resistant C2C12 cells, it is evident that PEG-BHD1028 effectively regulates glucose metabolism by lowering insulin resistance without stimulating insulin secretion. In conjunction with the results of activated cellular signal pathways responsible for glucose uptake, these data further imply that PEG-BHD1028 can be used to treat various diseases associated with adiponectin deficiency.

Besides the effect on the amelioration of insulin resistance, it has been widely known that adiponectin induces weight loss through fatty acid oxidation and energy expenditure. At the molecular level, adiponectin activates 5’ AMP-activated protein kinase (AMPK) and peroxisome proliferator-activated receptor alpha (PPARα) signal pathways [18,28]. AMPK pathway ultimately enhances mitochondrial biogenesis and fatty acid β-oxidation, and the activated PPARα promotes uptake, utilization, and catabolism of fatty acids by upregulating the genes involved in fatty acid transport, fatty acid-binding, and peroxisomal and mitochondrial fatty acid β-oxidation in the liver [32,33,34]. Interestingly, the animals received 5, 25, and 50 μg/Kg of the testing article showed significant weight loss in 21 days without compromising appetite. This result agrees with the results reported by Kubota et al. in 2007 [35]. Kubota’s study suggested that adiponectin induced appetite as an opposite action of leptin via activation of AMPK after binding to AdipoR1 and AdipoR2 in the hypothalamus. The inhibition of AMPK resulted in the suppression of the appetite-stimulating hormones, neuropeptide Y (NPY), and agouti-related protein.

As demonstrated in this study, PEG-BHD1028 significantly activated AMPK and subsequent signal pathways involved in fatty acid oxidation and mitochondrial biogenesis. These signal activations, in turn, led to a significant weight loss of the animals without affecting appetite in 21 days. Nevertheless, the physiological and biochemical mechanisms behind the delayed weight loss observed in most animals should be further investigated.

Although it has been confirmed that PEG-BHD1028 effectively activates signaling pathways involved in lipid metabolism and glucose metabolism in C2C12 cells and the type 2 diabetic animal model, the additional effects of PEG-BHD1028 on various metabolic disorders should be further evaluated. In particular, the liver constitutes an essential organ for lipid metabolism and accumulates excess fat by weakening the related metabolic functions. Representatively, non-alcoholic fatty liver disease (NAFLD) is closely associated with type 2 diabetes mellitus and obesity due to metabolic dysfunction such as insulin resistance and dyslipidemia and is also characterized by hepatic inflammation [36,37]. Other recent studies have shown that Alzheimer’s disease (AD) is also more likely to be affected by insulin resistance in the nervous system due to a metabolic syndrome condition [38]. In this regard, based on the known mechanisms of action of adiponectin in several diseases and the results of this study suggest that PEG-BHD1028 may provide a new therapeutic paradigm for metabolic disorders.

## 4. Materials and Methods

### 4.1. Materials

Dulbecco’s modified Eagle’s medium (DMEM), fetal bovine serum (FBS), and penicillin streptomycin (PS) were purchased from Hyclone (Logan, UT, USA), and Dulbecco’s Ca/Mg-free phosphate-buffered saline (DPBS) was from Welgene (Gyeongsan, Korea). Trypsin-EDTA, horse serum (HS), and trypan blue stain (0.4%) were from Gibco (Rockville, MD, USA), and human gAcrp30 (gAD, globular adiponectin) was from Prospec (East Brunswick NJ, USA). EzRIPA lysis buffer and West-Q Femto Clean ECL solution were purchased from Atto (Amherst, NY, USA), and BCA protein assay kit and PageRegular Plus Prestained Protein Ladder were from Thermo Scientific (Waltham, MA, USA). A 10% TGX Stain-Free Fastcast Acrylamide kit was from Bio-Rad (Hercules, CA, USA). Ammonium persulfate (APS) and bovine serum albumin (BSA) were purchased from Sigma-Aldrich (St. Louis, MO, USA) and TEMED from Bio basic (Markham ON, Canada). A 10X tris glycine electrophoresis buffer (with SDS), 10X tris glycine for native gel (without SDS), and 10X TBS-T were from HanLAB (Cheongju, Korea). Laemmli’s SDS-Sample buffer (4X, reducing) was from GenDEPOT (Hanam, Korea) and methyl alcohol (99.5%) from SAMCHUN (Seoul, Korea). A PVDF membrane was from Merck Millipore Ltd. (Burlington, MA, USA), and antibodies of Acetyl-CoA, Phospho-Acetyl-CoA, AMPKα, Phospho-AMPKα, PGC-1α, GAPDH, and anti-rabbit IgG were purchased from Cell Signaling Technology (Danvers, MA, USA). The mouse monoclonal antibodies MKP-1, PPARα, and anti-mouse IgG were purchased from SantaCruz Biotechnilogy, Inc. Skim milk was from Becton-Dickinson and Company (Seoul, Korea). Five weeks old male C57BLKS/J *db/db* mice were obtained from Japan SLC (Hamamatsu-shi, Japan). The plasma glucose concentration was measured using Accuchek Perfoma^®^ (Roche Diagnostics GmbH, Mannheim, Germany) and GDoctor^®^ (AGM-4000, All MEDICUS Co., LTD, Gyeonggi-do, Korea). The insulin concentration was measured using Morinaga UltraSensitive mice insulin ELISA Kit (Morinaga Institute of Biological Science, Yokohama, Japan) and SpectraMax M2 ELISA reader (Molecular Devices, San Jose, CA, USA). For the oral glucose tolerance test, D-(+)-Glucose from Sigma (St. Louis, MO, USA) was used. Animals were fed Purina Lab Rodent Chow 38057 (Pyeongteak, Korea).

### 4.2. Cell Culture and C2C12 Myotube Differentiation

Mouse C2C12 myoblasts (CRL-1772^TM^, ATCC, Manassas, VA, USA) were maintained in DMEM supplemented with 10% FBS and 1% PS at 37 °C in a 5% CO_2_ humidified incubator. When the confluence of the cells reached 80%, the culture medium was replaced with 2% horse serum (HS) containing DMEM to induce myotube differentiation, and the cells were incubated for 7 days with medium replacement every 24 h. The cells were then serum-starved overnight before PEG-BHD1028 or human gAcrp30 (gAD) treatment.

### 4.3. Cell Viability Assay

Cell viability was determined using an In vitro Toxicology Assay Kit (Sigma-Aldrich, St. Louis, MO, USA) based on MTT according to the manufacturer’s instructions. Briefly, the C2C12 myoblasts (3 × 10^4^ cells/well) were seeded in a 96-well plate and incubated for 24 h, then differentiated into myotubes for 7 days using 2% horse serum-containing DMEM. To examine the range of cytotoxicity induction of PEG-BHD1028, serial concentrations ranging from 7.8 nM to 1 mM were added into cell cultures for 24 h, then MTT labeling reagent solution (10 μL) was added to each well, and the solubilization solution was added after 3 h of incubation. The absorbance of the solubilized formazan was measured using an EPOCH microplate reader (BioTek Instruments, Inc., Winooski, VT, USA) at 570 nm.

### 4.4. Western Blot

Cells were lysed with an ice-cold lysis buffer using an EzRIPA lysis kit, and the protein concentration of the lysate was quantified using the BCA assay. Protein was mixed with Laemmli’s SDS-sample buffer and boiled for 5 min. An aliquot of the protein sample was loaded on SDS-PAGE and transferred onto a PVDF membrane. The membrane was blocked with 5% BSA for 1 h at room temperature on an orbital shaker followed by immunoblotting with the primary antibodies; ACC (1000:1), AMPKα (Thr172) (1000:1), PGC-1α (1000:1), p38 MAPK (1000:1), MKP-1 (500:1), PPARα (500:1), and GAPDH (2000:1), for overnight at 4 °C. The blotted membrane was washed 3 times with TBS-T at room temperature and reacted with horseradish peroxidase (HRP)-labeled anti-rabbit IgG (2500:1) or anti-mouse IgG (1000:1) according to the subject for 90 min at room temperature. Finally, bands were detected using a LuminoGraph II (Atto, Amherst, NY, USA) and quantified by densitometry using the CSAnalyzer4 (Atto, Amherst, NY, USA).

### 4.5. Evaluation of Mitochondrial Biogenesis

For the quantification of mitochondrial biogenesis, MitoTracker^TM^ Green FM (Invitrogen Life Technologies, Carlsbad, CA, USA) was used, which preferentially accumulates in mitochondria and provides an accurate quantitation of mitochondria. After appropriate treatments, cells were washed with PBS and incubated at 37 °C for 30 min with 100 nM MitoTracker^TM^ Green FM and stained in Hoechst33342 solution (Life Technologies, Eugene, OR, USA) sequentially. Fluorescence images and intensity was detected using an inverted Nikon A1Rsi Laser scanning Confocal Microscope with NIS elements software. Image acquisition and intensity calculations were conducted using CSAnalyzer software (ATTO, Tokyo, Japan). Briefly, the fluorescence intensity excluding the background value was normalized as a ratio to the number of cells in each region.

### 4.6. Quantitative Real-Time PCR Analysis for Mitochondrial Activities

Total RNA from cultured cells was extracted with the RNeasy mini kit (Qiagen, Hilden, Germany) according to the manufacturer’s protocol. The RNA concentration was measured using Qubit 4 Fluorometer (Life Technologies, Singapore), and single-strand cDNA was synthesized from 1 μg of RNAs using the High Capacity cDNA Reverse Transcription Kits (Applied Biosystems, Foster City, CA, USA) primed with a mixture of random primers. 10 times diluted cDNA template was used on the mixture of SYBR Green master mix (Applied Biosystems, Carlsbad, CA, USA) with 1 pmol of primers. The mouse-specific primer sequences (5 to 3 primes) for each gene are the following: COX2 forward, 5-TTT TCA GGC TTC ACC CTA GAT GA-3 and reverse, 5-GAA GAA TGT TAT GTT TAC TCC TAC GAA TAT G-3; ND5 forward, 5-TGG ATG ATG GTA CGG ACG AA-3 and reverse, 5-TGC GGT TAT AGA GGA TTG CTT GT-3; PGC-1α forward, 5-ACT ATG AAT CAA GCC ACT ACA GAC-3 and reverse, 5-TTC ATC CCT CTT GAG CCT TTC G-3; and Actin forward, 5-GGA AAA GAG CCT CAG GGC AT-3 and reverse, 5-GAA GAG TAT GAG CTG CCT GA-3 (Integrated DNA Technologies, Singapore). Quantitative PCR reactions were triplicated for each sample with QuantStudio 3 Real-Time PCR systems (Applied Biosystems, Carlsbad, CA, USA), and the threshold cycle (*C_T_*) for each reaction was normalized (Δ*C_T_*) by the value of the Actin housekeeping gene. Δ*C_T_* values were further normalized to compare the difference between the mean value of each sample.

### 4.7. Analysis of Mitochondrial DNA Copy Number

Mitochondrial DNA contents were determined using quantitative real-time PCR, as described previously [39]. Total DNA from cultured cells was extracted with the DNeasy Blood and Tissue Kit (Qiagen, Hombrechtikon, Switzerland) according to the manufacturer’s manual. The DNA concentration was measured in the EPOCH microplate reader using Take 3 Micro-Volume Plate (BioTek Instruments, Inc., Winooski, VT, USA). The template DNA was subjected at a concentration of 10 ng/mL with 1 pmol of primers for the quantitative PCR and the reaction was performed with SYBR Green using QuantStudio 3 Real-Time PCR systems (Applied Biosystems, Carlsbad, CA, USA). The mouse-specific primer sequences (5 to 3 primes) for each gene are the following: 16S rRNA (mitochondrial gene) forward, 5-CCG CAA GGG AAA GAT GAA AGA C-3 and reverse, 5-TCG TTT GGT TTC GGG GTT TC-3; ND1 (mitochondrial gene) forward, 5-CTA GCA GAA ACA AAC CGG GC-3 and reverse, 5-CCG GCT GCG TAT TCT ACG TT-3; and Hexokinase (nuclear gene) forward, 5-GCC AGC CTC TCC TGA TTT TAG TGT-3 and reverse, 5-GGG AAC ACA AAA GAC CTC TTC TGG-3 (Integrated DNA Technologies, Singapore). Quantification of mitochondrial DNA contents was performed by calculating the ratio of mitochondrial DNA copy number to nuclear DNA copy number, as described previously [39].

### 4.8. Surface Plasmon Resonance (SPR)-Based Binding Study

The intermolecular interaction between PEG-BHD1028 and AdipoRs was analyzed by SPR (Biacore T200, GE Healthcare (Sweden)). N-terminal truncated form of AdipoR1-Δ88 (residues 89–375) and AdipoR2-Δ99 (residues 100–386) were prepared as described in previous studies [12,40]. AdipoRs were immobilized at a level of 998.1 RU and 965.5 RU, respectively, in the sample channels on the surface of a CM5 sensor chip using amine coupling. For the immobilization procedure, AdipoRs were diluted to a final concentration of 15 μg/mL using a 10 mM Sodium Acetate buffer at pH5.0 and immobilized to the surface in separate flow cells at 10 μL/min. For kinetic experiments, PEG-BHD1028 was diluted in HBS-DT (10 mM HEPES, 150 mM NaCl, 5% (*v*/*v*) DMSO, pH 7.4) containing 0.005% (*v*/*v*) Tween 20 at concentrations ranging from 0.391 to 50 μM, and were injected at a flow rate of 30 μL /min over the immobilized AdipoRs. The data fit the kinetic model; Heterogeneous Ligand model using BiaEvaluation software V3.0 (Biacore, Piscataway, NJ, USA).

### 4.9. Measurement of Glucose Uptake

The 2-DG uptake of C2C12 cells was measured using the Glucose Uptake Assay Kit (Colorimetric, Ab136955, Abcam, Cambridge, UK) according to the manufacturer’s instruction. Palmitate to induce insulin resistance was prepared with minor modifications from the previous description [41]. Briefly, palmitate was dissolved in 0.1 M NaOH to a concentration of 20 mM by heating at 75 °C and the dissolved solution was then diluted with 5% fatty acid-free bovine serum at a stock solution of 6 mM at 55 °C. Cells were treated with 0.6 mM palmitate overnight. The fixed cells were washed three times with DPBS, followed by incubation in Krebs-Ringer-Phosphate-Hepes (KRPH) buffer containing 2% fatty acid free-BSA for 40 min. The cells were then further incubated in the presence or absence of 100 nM insulin and PEG-BHD1028 with the concentrations indicated in the figure for 30 min before the addition of 10 μL 2-DG (1 mM) in the 96-well plate at 37 °C. Absorbance was measured using an Epoch microplate reader (BioTek Instruments, Inc., Winooski, VT, USA) at 412 nm.

### 4.10. Animal Experiment

Upon arrival, C57BLKS/J *db/db* mice were examined for any physical or behavioral conditions and housed three animals per cage on a 12-h reverse-dark cycle (07:00–19:00 lighting at 150–300 lux). They were given 2 weeks to acclimate to their environment before being assigned to their respective experimental groups under room temperature between 21–23 °C with 45–60% humidity and maintained their *ad libitum* water and chow diet throughout the study period. PEG-BHD1028 solution was prepared in sterile PBS to give a concentration of 1 mg/mL as a stock solution for all in vivo studies. All procedures were carried out following the recommendations of the Korean Animal Protection Act and were approved by the Animal Care Committee at KPC. The IUCUC approval was obtained before each in vivo study: the approval numbers of the glucose-lowering effect and profiling, the OGTT, and the bodyweight studies were P172021, P182022, and P182024, respectively.

### 4.11. Glucose-Lowering Effect and Profile

The glucose-lowering effect and profile of PEG-BHD1028 in the *db/db* mouse model were evaluated. Six-week-old male animals were tested for their plasma glucose concentrations following the 6-h fasting to ensure the adequate onset of a diabetic phenotype. After two-week quarantine and adaptation, a total of 20 male mice were randomly divided into 4 groups (*n* = 5 per group), 1 control and 3 experimental groups, based on body weights and blood glucose concentrations measured a day before the study. All animals fasted for 6 h before the administration of the testing articles. The control and three experiment groups received a vehicle and a single dose of 50, 100, and 200 μg/Kg of PEG-BHD1028 subcutaneously, respectively, at a volume of 5 mL/Kg. The fasting blood glucose concentration was measured hourly for the first 8 h and every 2 h for the next 4 h by taking the sample from the catheterized tail vein. 

### 4.12. OGTT (Oral Glucose Tolerance Test)

For OGTT, 15 male *db/db* mice were randomly divided into 3 groups (*n* = 5 per group), 1 control and 2 experimental groups, based on the body weight and the fasting blood glucose measured a day before the test. The control and two experimental groups received a vehicle and a single dose of 1 and 50 μg/Kg of PEG-BHD1028 subcutaneously, respectively, at a volume of less than 5 mL/Kg at 240 min before glucose loading. For the test, 1 g/Kg of glucose was orally fed to animals (time point 0). Blood samples were collected through the catheterized tail vein, and the plasma glucose and insulin concentrations were measured at −240, 0, 15, 30, 60, and 120 min.

### 4.13. Body Weight and Food Intake

To evaluate the effects of PEG-BHD1028 on body weight and food intake, 30 male *db/db* mice were randomly divided into 6 groups (*n* = 5 per group), 1 control, and 5 experimental groups, based on body weights and fasting glucose concentrations measured a day before the experiment. The control and five experiment groups received a vehicle and 1, 5, 25, 50, and 200 µg/Kg of PEG-BHD1028 daily for 21 days, respectively, at a volume of 5 mL/Kg via a subcutaneous injection. The change in body weight and food intake was monitored for 21 days along with other clinical observations.

### 4.14. Statistical Analysis

The variability of results was expressed as the mean ± standard error of the mean (SEM). Statistical analyses were carried out using GraphPad Prism 5.0 (GraphPad Software, La Jolla, CA, USA). The significance of differences was determined using either an unpaired two-tailed Student’s *t*-test or a paired one-tailed Student’s *t*-test as appropriate. A *p*-value of ≤ 0.05 was considered to indicate a statistically significant difference.

## 5. Conclusions

PEG-BHD1028 is a pegylated peptide consisted of 15 amino acids and designed to be an agonist to adipoR1 and R2 receptors with a high affinity. This study was performed to evaluate the function of the molecule against scientific information on adiponectin focusing on the mechanism of action using in vitro and in vivo models. The results of this study indicated that PEG-BHD1028 exhibited adiponectin-like activities in both testing models, they also imply that PEG-BHD1028 can be further developed into a therapeutic agent for various medically unmet diseases associated with adiponectin deficiency such as type 2 diabetes mellitus, obesity, non-alcoholic steatohepatitis (NASH), and Alzheimer’s disease.

## Figures and Tables

**Figure 1 ijms-22-00884-f001:**
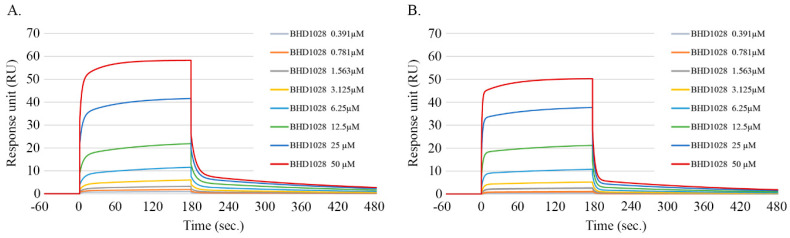
Representative sensorgrams of the BHD1028-AdipoRs binding overlay plot with the kinetic fit. The BHD1028-receptor binding sensorgrams were obtained after injection of various concentrations of BHD1028 into an AdipoR1-Δ88 and AdipoR2-Δ99 on the immobilized CM5 chip at pH 5.0. Concentrations titrated at a 1:1 ratio with red being the highest concentration and blue the lowest concentration flowing at 30 μL/min with a total association time of 180 s and disassociation of 300 s. The data were presented in types of the kinetic model: The equilibrium dissociation constant at the heterogeneous ligand model ((**A**) AdipoR1-BHD1028. (**B**) AdipoR2-BHD1028). SPR signals from different concentrations were plotted against concentrations to calculate a Kd value.

**Figure 2 ijms-22-00884-f002:**
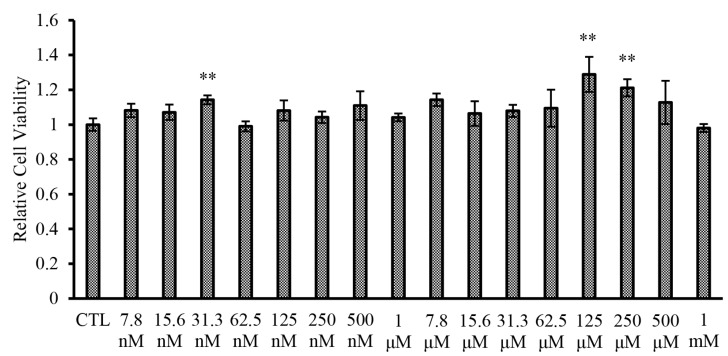
Effect of PEG-BHD1028 on cytotoxicity in C2C12 myotubes. C2C12 myotubes were treated with the serial concentrations of PEG-BHD1028 ranging from 7.8 nM to 1 mM for 24 h. The results are shown as a mean ± standards error of the mean (*n* = 3) relative to the control samples. ** *p* ≤ 0.01 vs. CLT (Negative control).

**Figure 3 ijms-22-00884-f003:**
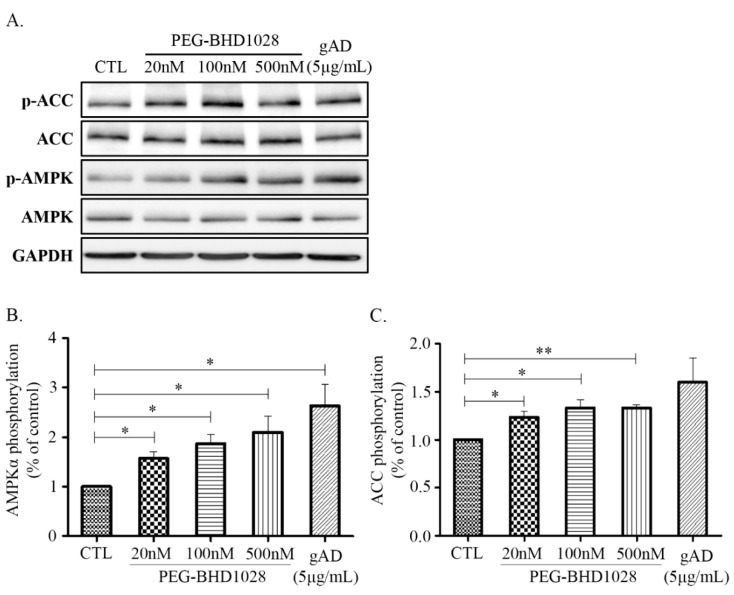
Immunoblot analysis of AMPK and ACC phosphorylation in C2C12 muscle cells. Differentiated C2C12 myotubes were treated with 20, 100, 500 nM of PEG-BHD1028 and globular adiponectin (5 μg/mL) was treated as a positive control for 30 min. The phosphorylation of AMPK and ACC was analyzed by western blot (**A**). Western blot signals were quantified by a densitometer (**B**,**C**). Results are presented as means ± standards error of the mean (*n* = 3). ** *p* ≤ 0.01, * *p* ≤ 0.05 vs. CTL (Negative control).

**Figure 4 ijms-22-00884-f004:**
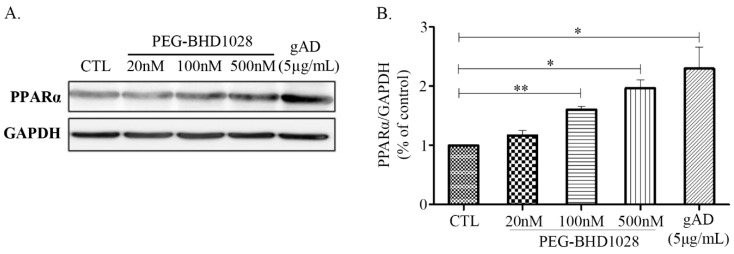
The relative protein levels of PPARα activity in differentiated C2C12 myotubes. (**A**) Representative blots illustrating the effect of PEG-BHD1028 on PPARα activation. (**B**) Western blot signals were quantified by a densitometer. The amount of protein was normalized to GAPDH. Results are presented as means ± SEM (*n* = 3). ** *p* ≤ 0.01, * *p* ≤ 0.05 vs. CTL (Negative control).

**Figure 5 ijms-22-00884-f005:**
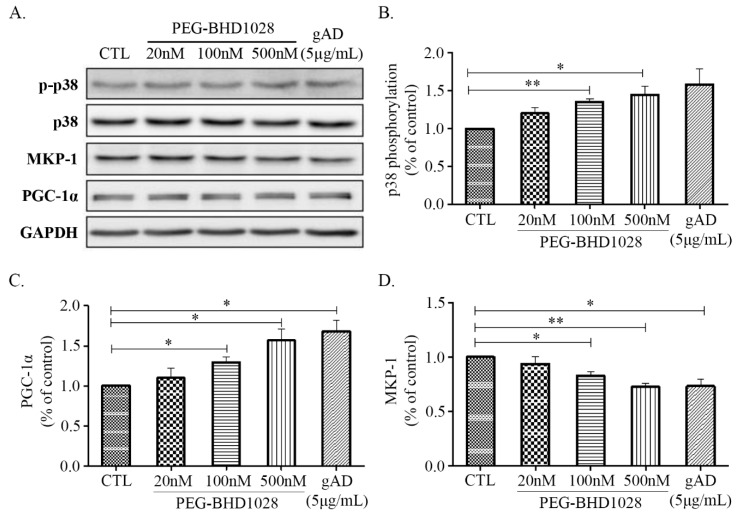
The regulatory effects of PEG-BHD1028 on mitochondrial biogenesis in C2C12 myotubes. (**A**) The activation of p38 MAPK, PGC-1α, and MKP-1 following the treatment with PEG-BHD1028 was observed by western blot. (**B**) The change in the ratio of the phosphorylation band density of p38 MAPK to control MAPK was quantified. (**C**,**D**) The quantification of PGC-1α and MKP-1 proteins was represented as a ratio to GAPDH. Results are presented as means ± SEM (*n* = 3). ** *p* ≤ 0.01, * *p* ≤ 0.05 vs. CTL (Negative control).

**Figure 6 ijms-22-00884-f006:**
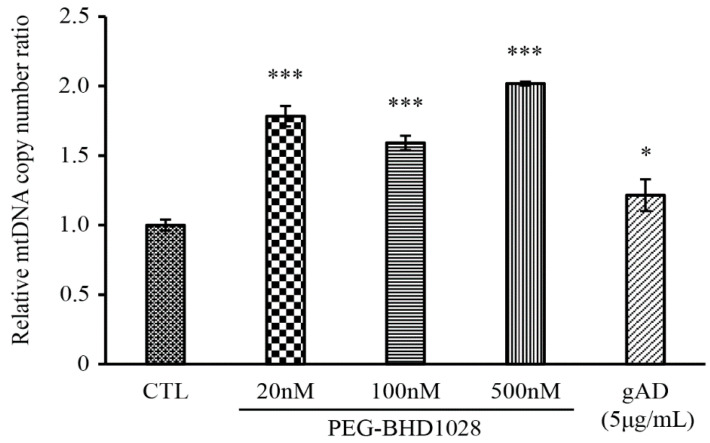
The DNA copy number of mitochondria specific gene, ND1 was quantified using qPCR. The relative copy number of mitochondrial DNA was presented as a ratio to the copy number of nuclear specific DNA, hexokinase (HK). Results are presents as means ± SEM (*n* = 3). * *p* ≤ 0.05, *** *p* ≤ 0.0001 vs. CTL (Negative control).

**Figure 7 ijms-22-00884-f007:**
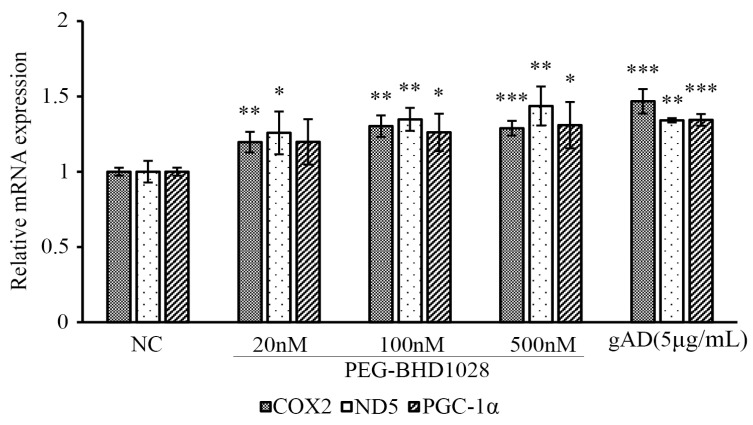
PEG-BHD1028 increases mitochondria-related gene expression in C2C12 myotubes. mRNA expressions of COX2, ND5 and PGC-1α were quantified using qPCR in C2C12 myotubes. The threshold cycle (*C_t_*) for each reaction was normalized (Δ*C_t_*) by the value of the β-Actin. The value of Δ*C_t_* was further normalized to exhibit the comparative expression levels with respect to the mean value. Results are presented as means ± SEM (*n* = 3). *** *p* ≤ 0.001, ** *p* ≤ 0.01, * *p* ≤ 0.05 vs. Negative control.

**Figure 8 ijms-22-00884-f008:**
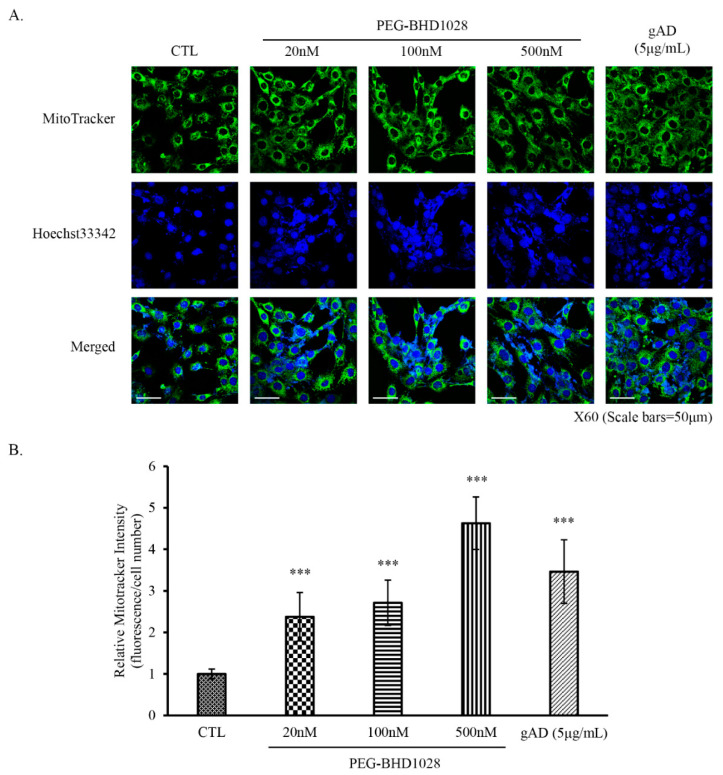
PEG-BHD1028 increases mitochondrial biogenesis in C2C12 myotubes. C2C12 myotubes were treated with 20, 100, and 500 nM of PEG-BHD1028 and 5 μg/mL of gAD for 24 h. Mitotracker-stained mitochondrial intensity of cells (**A**) was measured using a confocal fluorescence microscope, and the relative mitotracker intensity against nuclear DAPI intensity (**B**) was calculated. Each fluorescence intensity excluding background of 5 spots of 82 mm^2^ in the same area was obtained. The intensity of each cell was normalized reflecting the confluence coefficient. Results are presents as means ± SEM (*n* = 5). *** *p* ≤ 0.0001 vs. CTL (Negative control).

**Figure 9 ijms-22-00884-f009:**
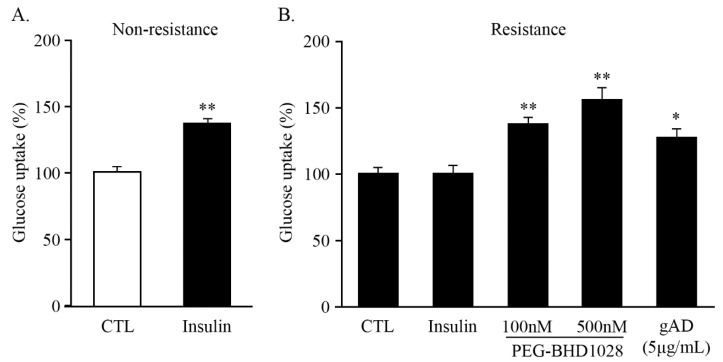
Glucose uptake enhancement effect of PEG-BHD1028. Effect of insulin on the uptake of 2-deoxy-D-glucose (2DG) in non-resistance (**A**). PEG-BHD1028 ameliorated palmitate-induced insulin resistance in C2C12 myotubes (**B**). Globular adiponectin (5 μg/mL) was treated as a positive control. Results are presented as means ± standards error of the mean (*n* = 5). ** *p* ≤ 0.01, * *p* ≤ 0.05 vs. CTL (Negative control).

**Figure 10 ijms-22-00884-f010:**
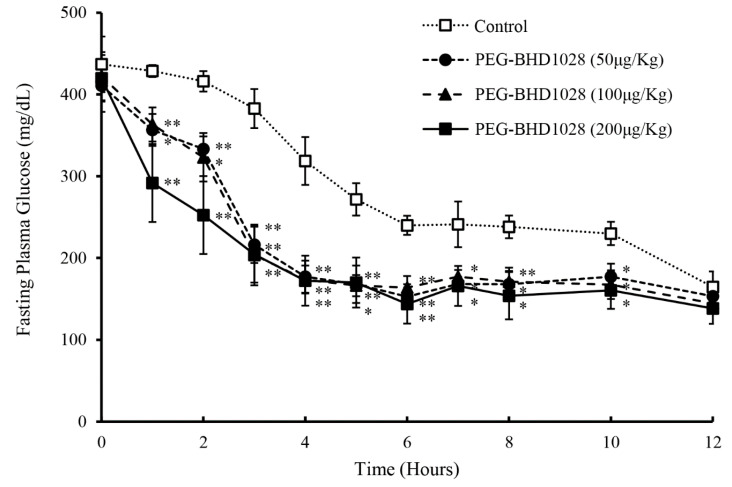
Glucose-lowering effect and pharmacodynamic profile of PEG-BHD1028 following s.c. administration of 50, 100, and 200 μg/Kg PEG-BHD1028 in *db/db* mice (*n* = 5/group). ** *p* ≤ 0.01, * *p* ≤ 0.05 vs. control.

**Figure 11 ijms-22-00884-f011:**
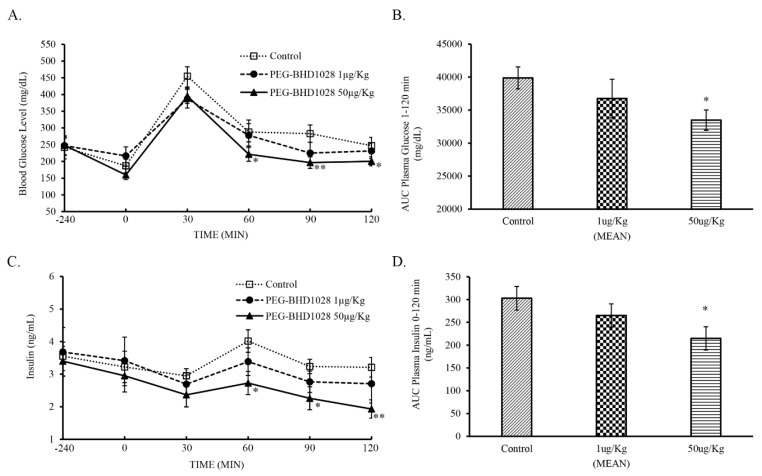
Oral glucose tolerance test following the s.c. administration of 1 and 50 μg/Kg of PEG-BHD1028. Effect on blood glucose (**A**), blood glucose AUC from 0 to 120 min (**B**), effect on plasma insulin (**C**), and insulin AUC from 0 to 120 min (**D**). Each data point in (**A**,**C**) and each bar in (**C**,**D**) represent the mean of 5 animals ±SE. ** *p* ≤ 0.01, * *p* ≤ 0.05 vs. control.

**Figure 12 ijms-22-00884-f012:**
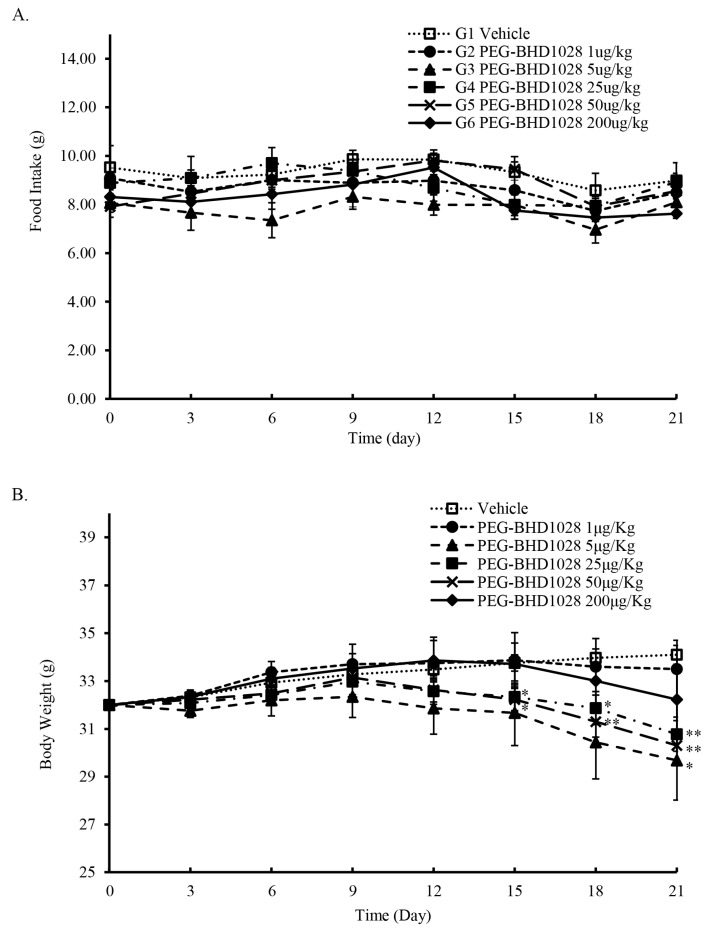
Effect of PEG-BHD1028 on food intake and weight loss. (**A**) The amount of food intake after the administration of PEG-BHD1028 was relatively consistent during the testing period. (**B**) Bodyweight of the 25 and 50 μg/Kg groups were significantly decreased compared to the control groups from day 15 (*p* ≤ 0.05 and *p* ≤ 0.01 on day 21) and 5 μg/Kg group from day 21 (*p* ≤ 0.05). Each point represents the mean of five animals ± S.E. ** *p* ≤ 0.01, * *p* ≤ 0.05 vs. control.

**Figure 13 ijms-22-00884-f013:**
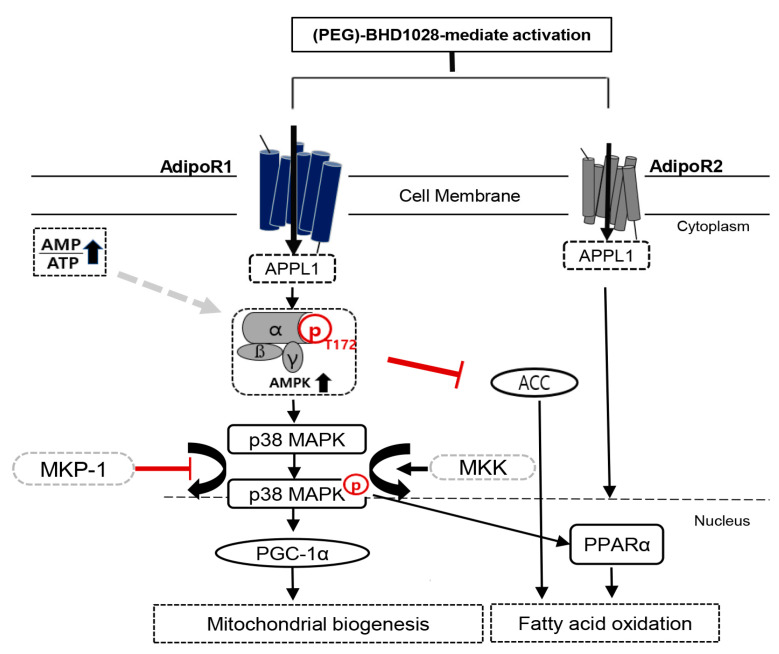
Schematic diagram of the PEG-BHD1028 signaling for mitochondrial biogenesis and fatty acid β-oxidation in muscle cells. The signal activated by PEG-BHD1028 positively regulates energy expenditure and lipid metabolism, primarily through downstream activation of AMPK after binding to AdipoR1 via the mediation of the adaptor protein phosphotyrosine interaction (APPL1). The phosphorylated AMPK activates p38 MAPK by inhibiting the MKP-1 expression, resulting in the activation of PGC-1α to promote mitochondrial biogenesis. Activated AMPK also enhances fatty acid oxidation by phosphorylating ACC. Fatty acid β-oxidation can be induced by activation of PPARα after PEG-BHD1028 binds to AdipoR2.

**Table 1 ijms-22-00884-t001:** SPR analysis of binding affinity of BHD1028 to AdipoR1 and AdipoR2.

Parameter	Binding Site 1			Binding Site 2		
	Ka1 (M^−1^s^−1^)	Kd1 (1/s)	KD1 (µM)	Ka2 (M^−1^s^−1^)	Kd2 (1/s)	KD2 (µM)
AdipoR1	502.9	0.003713	7.38	3633	0.1328	36.55
AdipoR2	351.3	0.003956	11.26	10610	0.2659	25.06

Ka: Association constant. Kd: Dissociation constant. KD: Equilibrium dissociation rate constant (Kd/Ka).

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
