# Peer review of "PEG-BHD1028 Peptide Regulates Insulin Resistance and Fatty Acid β-Oxidation, and Mitochondrial Biogenesis by Binding to Two Heterogeneous Binding Sites of Adiponectin Receptors, AdipoR1 and AdipoR2"

_ijms, 2021, doi:10.3390/ijms22020884_

Round 1

Reviewer 1 Report

First and the foremost, this article is in part, reiteration of the previous study by the same team of authors that has been published two years ago on PLoS One (Kim et al., 2018) Namely, the binding affinities towards AdipoR1-Δ88 by Surface Plasmon Resonance as well as AMPK activation in C2C12 myotubes by of BHD1028 have been already studied and the results presented.

The title is too vague and quite misleading. What does “alternative driver” mean in this context. Needs to reflect the specific results only, obtained in the study. 

What does the hieroglyph character for “-oxidation” in the keywords mean? 

Introduction is exceedingly long and sounds more like a review. In fact, this is a brief report on biological characterization of some adiponectin-mimicking peptide (BHD1028) and its PEGylated derivative. The lines 35-59 should be shortened to just 1-2 sentences and mostly focused on adiponectin peptide mimics previously described in the literature, their characteristics, importance of introducing this new one etc. Instead, a slightly more detailed explanation of how the precise peptide sequences denoted has been selected. 

Peptide structure, with molecular weights of (non-) and PEGylated versions and the binding sites for A highlighted, is not provided. 

This poses another question regarding selected concentration (5 μg/mL) for globular adiponectin (gAD) used as a positive control. Is it equimolar to 20, 100, or 500 nM of PEG-BHD1028? 

Do AdipoR1-Δ88 and AdipoR2-Δ99 correspond to the chip-immobilized peptide sequences or to the mutant proteins, with modified AdipoR2 and AdipoR1 regions, correspondingly? 

Why authors characterize the binding affinity of non-PEGylated, while most of the other assays are done on PEG-BHD1028? How these have been considered to be relevant? Have the self-assembly capacities of PEGylated molecules been evaluated (e.g. by DLS). The larger aggregates may have totally different in vitro and in vivo characteristics, not even speaking about binding affinities. 

While for elevated p38 MAPK expression there are no questions, changes in PGC-1α levels appear to be totally indistinguishable from WB gel. Would be helpful to see the original data of intensity measured, from which Fig. 4C has been drawn. Also, a substantial increase in total DNA content of NADH dehydrogenase subunit for peptide but not gAD is indicative of a different mechanisms of action by the peptide supposedly, arising from the non-specific cellular stress. Have the authors measured the cytotoxicity by PEG-BHD1028 to exclude or confirm this? 

In Fig. 7B it is not entirely clear how authors derived the relative fluorescence intensities. Per square μm? Per single cell? Per micrograph frame? In Fig. 7A the cell confluence for control appears to be lower apparently, resulting in less intensity per the same area. 

Author Response

Dear reviewer,

Reviewer 2 Report

The presented research results are very valuable and well-conducted. The work was prepared very carefully with attention to detail.

The authors have already described the properties of the peptide PEG-BHD1028 in their previous paper. However, the research data presented in this paper mainly concentrated on the influence of PEG-BHD1028, a potent novel peptide agonist to AdipoRs, on activation AMPK and subsequent pathways and enhancing fatty acid β-oxidation and mitochondrial biogenesis.

Question: It is expected that more details should be placed in the manuscript about this peptide. The structure of the peptide,  and information about cytotoxicity.

The authors showed that PEG-BHD1028 facilitated glucose uptake by lowering insulin resistance in insulin-resistant C2C12 cells. Besides, PEG-BHD1028 significantly lowered the fasting blood plasma glucose level in db/db mice following a single s.c. 20 injection of 50, 100, and 200 μg/Kg and glucose tolerance at a dose of 50 μg/Kg with significantly decreased insulin production. The animals received 5, 25, and 50  μg/Kg of PEG-BHD1028 for 21 days significantly lost their weight after 18 days in a range of 5% - 7%. These results imply the development of PEG-BHD1028 as a potential adiponectin replacement therapeutic agent.

Question: There is no information in the paper why such doses of the peptide PEG-BHD1028 were proposed in each experiment? Have they been experimentally established? Have any available compounds having a similar effect had suggested the authors' proposed doses? What is the reason for the different numbers of mice in each study?

For future: In subsequent studies, computational studies to demonstrate the binding site will be very valuable.

Author Response

Dear reviewer,

Round 2

Reviewer 1 Report

1. I thank the authors for a detailed explanation on the initial screening of peptide candidates vs. the characterization of the selected one. As a continuation of their previous study, the repetitive results by WB and SPR should be placed in the supplementary material while keeping the reference to the original publication. In turn, the cytotoxicity data are quite important and thus, need to be presented in the manuscript text. 

2. I still believe the title should be very specific and based solely on the results obtained, rather than on speculations made. I understand it makes the content more clickbait and eye-catching, yet not appropriate for research articles. Although PEG-BHD1028 displays somewhat similar effect on AMPK/ACC signalling pathway, metabolic oxidation, and glucose uptake in vivo, a substantial difference in amplification and expression of mitochondria-associated genes does not support the mechanism of action, same as for adiponectin. It is also still prematurely to propose the usability of peptide to substitute adiponectin, in this case. 
As an example, below are two optional titles provided, which I could both agree with. 

PEGylated BHD1028 peptide regulates fatty acid oxidation and mitochondrial biogenesis by binding to two heterogeneous binding sites on adiponectin receptor” 

The role of PEGylated BHD1028 peptide in lipid and glucose metabolism by interaction with adiponectin receptor” 

3-7. Mainly resolved and/or explained. No further objections. 

8. See p. 2. 

9. Understood. In this case, the authors need to adjust the intensity values shown in in Fig. 7B by the cell confluence coefficient (i.e., more cells per frame results in higher overall intensity to be considered). 

In summary, the following needs to be done to make the manuscript publishable: 

  • Distribute the data between manuscript text and supporting information properly (p. 1) 
  • Change title to be less speculative (p. 2) 
  • Normalize fluorescent intensities over the cell density within the monolayer (p. 9) 

Author Response

Dear,reviewer

Reviewer 2 Report

The manuscript was corrected according to the reviewer suggestion. I accept the manuscript in present form.

Author Response

Dear reviewer,

We highly appreciate  your  comments and suggestions.

Sincerely,

EncuraGen, Inc.